# OpenReview forum: "Is GPU Numerical Noise Really Random? An Empirical Investigation of Floating-Point Error Structure"
_ICLR.cc/2026/Workshop/Sci4DL — Sci4DL 2026_

### Official Review · Reviewer_8FKV · 2026-02-23

**Fit:** 3
**Significance:** 2
**Confidence:** 3

**Summary:**

The paper looks at numerical precision issues using 2 forms of 16-bit floating point number. it finds that the distribution of errors is not i.i.d, but rather has some structure.

**Strengths:**

While brief, the paper presents an easily understandable and probably important finding in how DL works, particularly in understanding the impact of number representation. As noted below, this topic has been of great interest in the early days of DSP, when a lot of research was done. Suggest you check text books by Oppenheim & Schafer and Rabiner & Gold.

**Suggestions:**

The signal processing community did a lot of research on a very similar topic in the 70s and 80s. It's worth checking this out so as not to re-invent the wheel. While none of that work looked a bfloat, there was research both on 8 bit integers and on 16-bit float with similar exponent to IEEE - that standard came later.

i would be interesting to see the pdfs etc. and any other visualisations that go beyond the terse tables of results presented.

please either explain what you mean by 'static' or, if you use it as a synonym for noise, don't use it. it gives the impression that you mean stationary, as in stationary white noise, but i don't think you mean that.

---

### Official Review · Reviewer_CgSh · 2026-02-27

**Fit:** 2
**Significance:** 2
**Confidence:** 1

**Summary:**

The authors study the non-determinism of matrix multiplication on GPUs. They show that a (potential) assumption that the noise introduced is zero-mean Gaussian and i.i.d does not hold, when empirically measured using a single matrix multiplication setup, sampled 10k times. Under these assumptions, the model predicts a certain logit ‘flip rate’ across trials, but none are observed - implying that GPU noise is not random but structured. Implications for GPU-trained deep models (possibly being more noise-stable than expected) are conjectured but not explored.

**Strengths:**

This is an interesting topic to tackle, and the work is clearly presented and concisely written. The results are interesting, and the correlated noise observed is notable. The logic is sound and the work seems technically correct.

**Suggestions:**

The paper is still very light, even for a workshop paper. The background references mentioned do not seem to address GPU determinism specifically, so I am unsure how important the Gaussian i.i.d assumption is here - consider substantiating and explaining this more clearly. It would also have been useful to repeat the experiment in more settings (different number of dimensions, or with more than one hardware setup) to determine sensitivity to such settings.

[Note to Meta-reviewer: I was somewhat on the fence between selecting a 1 or a 2 for significance.]

Some minor detail:
- What is the reason for selecting \tau as 5?
- In Section 2.2, consider pointing the reader to the derivations in Appendix A.
- Consider including the visualisation referred to in the “Visualisation Insight” in the appendix.
- Please introduce symbols and abbreviations, even if fairly obvious.

---

### Meta-Review · Area_Chair_jAPV · 2026-03-02

**Recommendation:** Accept

**Metareview:**

This work studies the non-determinism of matrix multiplication on GPUs and shows that the widespread assumption that the noise introduced is zero-mean Gaussian and i.i.d does not hold. Instead, the noise is structured, resulting in avoiding logit flips. Although the paper is very short, the reported phenomenon can feed into an interesting discussion.

---

### Decision · Program_Chairs · 2026-03-02

Accept